# Secretome of Adipose-Derived Stem Cells Cultured in Platelet Lysate Improves Migration and Viability of Keratinocytes

**DOI:** 10.3390/ijms24043522

**Published:** 2023-02-09

**Authors:** Maike Hermann, Ajay Peddi, Alexander Gerhards, Rafael Schmid, Deborah Schmitz, Andreas Arkudas, Volker Weisbach, Raymund E. Horch, Annika Kengelbach-Weigand

**Affiliations:** 1Laboratory of Tissue Engineering and Regenerative Medicine, Department of Plastic and Hand Surgery, University Hospital of Erlangen, Friedrich-Alexander-Universität Erlangen-Nürnberg, 91054 Erlangen, Germany; 2Department of Plastic Surgery, Hand and Burn Surgery, University Hospital RWTH Aachen, 52074 Aachen, Germany; 3Department of Transfusion Medicine and Haemostaseology, University Hospital of Erlangen, Friedrich-Alexander-Universität Erlangen-Nürnberg, 91054 Erlangen, Germany

**Keywords:** adipose-derived stem cells, platelet lysate, cell differentiation, cell culture, regenerative medicine, wound healing therapy, personalized medicine, molecular mechanism

## Abstract

Chronic wounds depict a silent epidemic challenging medical professionals worldwide. Regenerative medicine uses adipose-derived stem cells (ADSC) in promising new therapies. In this study, platelet lysate (PL) as a xenogen-free substitute for foetal bovine serum (FBS) in ADSC culture was used to create an ADSC secretome containing cytokines for optimal wound healing conditions. The ADSC secretome was tested on keratinocytes for migrational behaviour and viability. Therefore, human ADSC were characterized under FBS (10%) and PL (5% and 10%) substitution, regarding morphology, differentiation, viability, gene and protein expression. ADSC were then cultured in 5% PL and their secretome was used for stimulation of keratinocyte migration and viability. To enhance the effect, ADSC were treated with Epithelial Growth Factor (EGF, 100 ng/mL) and hypoxia (1% O₂). In both PL and FBS groups, ADSC expressed typical stem cell markers. PL induced a significantly higher increase in cell viability compared to FBS substitution. ADSC secretome contained various beneficial proteins which enhance the wound healing capacity of keratinocytes. This could be optimized treating ADSC with hypoxia and EGF. In conclusion, the study shows that ADSC cultivated in 5% PL can effectively support wound healing conditions and can be considered as a promising new therapy for individual treatment of chronic wound disorders.

## 1. Introduction

Chronic wounds represent a common problem amongst patients, therapists, and the health care system. Individual factors, such as local tissue hypoxia, bacterial colonization and decreasing stress answer on cellular and systematic levels of ageing patients, result in a long-lasting non-healing tendency of anatomic and physiological skin layers [1]. Thus, leading to an increase in pain, psychological, financial, and functional strains, and prolonged hospitalization [2]. Often underestimated, chronic wounds depict a silent epidemic. Combined with a rising prevalence of diabetic, venous, and arterial insufficiency ulcers and an ageing population, medical care is facing a large fraction of affected patients [3]. While the international gold standard of treatment includes topical care, debridement and skin grafts, clinical trials include novel therapies, such as application of growth factors, biological hydrogels and hyperbaric and low-pressure therapies [4].

The process of tissue defect healing is generally divided into three major phases: inflammation, proliferation and remodelling, which involves fibroblasts, keratinocytes, endothelial cells and platelets directed by para- and autocrine mechanisms [5]. Multipotent mesenchymal stem cells (MSC) are recently widely used in regenerative medicine, playing a key role in skin regeneration. Not only their ability to differentiate into various lineages, but also their secretome including regulatory molecules and trophic mediators, is beneficial for wound healing [6,7]. Extracellular vesicles are released from endosomal compartments, and contain a mixture of miRNA, mRNA and growth factors, that orchestrate angiogenesis, immunomodulation and tissue remodelling [8]. Adipose tissue is an easily accessible and economically advantageous source of adipose-derived stem cells (ADSC), which are mesenchymal stem cells, and can be used to create a conditioned medium, containing multiple proteins, such as Pentraxin-related Protein 3 (PTX3), Interleukin-8 (IL-8), Monocyte Chemotactic Protein 1 (MCP-1), Fibroblast Growth Factor 7 (FGF7), Heparin Binding Epidermal Growth Factor-Like Factor (HB-EGF) and Vascular Endothelial Growth Factor (VEGF) [9] known to enhance dermal and epidermal healing processes.

Platelets play a key role in wound healing, not only as active wound closers, but also as a source of growth factors, e.g., Platelet-derived Growth Factor (PDGF), Transforming Growth Factor Beta (TGF-β), Epidermal Growth Factor (EGF), FGF, VEGF und Connective Tissue Growth Factor (CTGF), and modulators of immune response. Platelet Rich Plasma (PRP), Platelet Rich Fibrinogen (PRF) or Human Platelet Lysate (PL) contain these factors and therefore could be beneficial for tissue engineering experiments [2]. The commonly used Fetal Bovine Serum (FBS) as media supplement for ADSC is economic and practical but has shown to have an increased risk of contamination and immunization, has a high lot-to-lot variability and contains xenogeneic components, which are possible triggers for allergic reactions and carriers of viruses [10]. PL has shown to be a safe alternative for FBS [11,12] without xenogeneic-related complications. Preliminary studies came to different perceptions regarding the effects of PRP but highlighted the necessity of further investigation and superior randomly controlled studies for evidence and efficacy of PRP [13]. Through multiple freezing, defrosting and centrifugation cycles, PL can be produced in a simple, inexpensive, and fast way from PRP. This procedure destroys the membrane of platelets and elevates the concentration of secretory factors [14].

Physiologically, wounds represent a low oxygen environment, hence a low oxygen incubation of ADSC cultures could lead to an accelerated amount of growth factors. Secretome of ADSC cultured in 1% and 5% oxygen was analysed, showing a higher amount of anti-apoptotic and angiogenic cytokines [15]. In other studies, hypoxia was combined with EGF substitution [16], leading to an increased number of secretory factors. EGF can enhance cell proliferation and differentiation [17,18,19]. For optimized wound healing, the combination of hypoxia and EGF could be a benefit for maximal yield of secretory factors necessary for wound healing.

Within this study, ADSC were cultivated in PL and effect on cell differentiation, viability, and expression of surface markers were analysed and compared to conventional cell culture conditions with FBS. Different types of ADSC secretome cultivated in PL under normoxia or hypoxia with or without EGF addition were produced and their influence on keratinocytes was analysed, thus mimicking physiological wound healing in vitro. The aim of the present study was to examine the efficacy of ADSC secretome from PL cultivated ADSC as possible therapeutic approach for chronic wound disorders, which would be easily accessible, storable, and cheap.

## 2. Results

### 2.1. PL Maintains Typical ADSC Morphology, Immunophenotype and Differentiation Pattern

From all donors, ADSC could successfully be isolated from abdominal fat tissue and cultivated at least over eight passages.

#### 2.1.1. ADSC Morphology

The morphology of ADSC in different experimental groups was checked after 24 h and 72 h. Cell confluency was reached much faster in groups cultivated in PL. In all groups, ADSC showed the typical spindle-shaped, fibroblast-like morphology and grew homogenously distributed on the well plates (Figure 1a). No signs of apoptosis such as blebbing were visible.

#### 2.1.2. Increase in ADSC Viability and Proliferation Cultured in Platelet Lysate

The effect of PL on metabolic activity was quantified by viability assay WST-8 over the course of 72 h (Figure 1b). Both PL groups showed a statistically significantly increased cell viability compared to FBS groups after 48 h and 72 h. Mean values of absorbance of ADSC in PL were nearly twice as high as those of ADSC grown in FBS. There was no significant difference of cell viability between 5% and 10% PL groups. Individual values of ADSC donors regarding viability are shown in Appendix A. For comparison of cell proliferation between the experimental groups, gene expression of the proliferation marker *MKI67* was analysed by RT-PCR. It could be shown that *MKI67* expression in 5% and 10% PL groups was 3.8 times and 5.3 times higher compared to FBS group (Figure 1c).

#### 2.1.3. Stem Cell Markers

For further characterization of ADSC in different experimental groups, typical stem cell surface marker expression was analysed by flow cytometry. Overall, it could be shown, that all characteristic markers were expressed in all groups without statistical differences except for CD105. CD105 in PL groups showed significantly lower expression, though still very high. All groups strongly expressed positive markers such as CD73, CD90 and CD105, while only very low expression of negative markers CD11b, CD19, CD31, CD34, CD45 and HLA-DR (Lin-cocktail) could be detected (<2%) (Figure 2).

#### 2.1.4. Differentiation

ADSC were differentiated into the adipogenic and chondrogenic cell lineage to analyse the effect of PL on their differentiation potential.

Adipogenic differentiated ADSC were analysed by microscopy and Oil Red O staining after day 21. In all groups, ADSC could successfully be differentiated. ADSC in PL developed earlier signs of adipogenic differentiation such as intracellular lipid droplets around day 7 of induction. The amount and size of lipid droplets increased rapidly over time. However, due to the high proliferation of ADSC in PL groups, a time point of 21 days could not be analysed due to detachment of complete cell layers and only FBS groups were stained with Oil Red O (Figure 3a).

Chondrogenic differentiation capability was proven after 4 weeks of chondrogenic differentiation. In all groups, alcian blue staining confirmed the expression of chondrogenic proteins, such as glycosaminoglycans without visible difference between the experimental groups (Figure 3b).

### 2.2. Platelet Lysate Induces the Secretion of Growth Factors in ADSC

In order to analyse the secretome of ADSC in different groups, a variety of proteins were examined by ELISA Quantibody (*n* = 5). A range of cytokines important for wound healing (HGF, VEGF, VEGFR2, EG-VEGF, EGFR, FGF-4, MCSF, MCSFR, SCF and SCFR) were highly secreted in PL groups compared to the FBS group (Figure 4), where VEGF, MCSF, MSCFR, SCF were statistically significant and EGFR and FGF 4 were highly significant. The HGF concentration was 3.8 times higher in 5% PL and 2.9 times higher in 10% PL in comparison to the FBS group. VEGF-R2 concentration was highest in 5% PL. All other cytokines, VEGF, EG-VEGF, EGFR, FGF-4, MCSF, MCSFR, SCF and SCFR showed the highest levels in the 10% PL group, up to four times higher compared to the FBS group. ICAM-1 and RANTES were significantly increased in PL groups. MCP-1 is also elevated in PL groups. No significant differences were detected between 5% and 10% PL groups. A high patient-to-patient variability was observed.

### 2.3. Immunofluorescence Staining of Human Primary Keratinocytes

Isolated keratinocytes were characterized using immunofluorescence staining. In total, 100% of stained cells were positive for pan-cytokeratin (monoclonal mouse anti-human cytokeratin clone MNF116) (Figure 5).

### 2.4. Secretome of EGF and Hypoxia Pretreated ADSC Effects Keratinocyte Viability and Migration

Keratinocytes were then cultivated in ADSC secretome from ADSC cultivated in 5% PL in either normoxia or hypoxia with or without supplementation of EGF. The effect on cell viability and migration was analysed over 7 days, and 24 h, respectively (Figure 6). In all groups, keratinocytes migrated over time. In hypoxia groups, EGF significantly stimulated keratinocyte migration after 24 h (Figure 6a). There was further a tendency for a higher migratory activity in hypoxia groups compared to the normoxia group.

In normoxia groups, EGF significantly stimulated keratinocyte viability after 4 and 7 days (Figure 6b). Likewise, in hypoxia groups, EGF stimulated keratinocyte viability highly significantly after 4 d. After seven days, the normoxia EGF group showed an about 2-fold higher absorbance compared to the normoxia group. The combination of hypoxia and EGF induced a significantly higher viability compared to the normoxia group after 4 days. Interestingly, in EGF groups, hypoxia did not further increase the cell viability compared to normoxia. No statistical differences between normoxia and hypoxia groups without EGF could be observed.

## 3. Discussion

PL is currently used in various studies in the field of regenerative medicine [20,21,22,23,24]. In this study, first we proved that PL is a superior, safe and easily available substitute for FBS in ADSC cell culture. Secondly, we optimized ADSC secretome for further clinical usage since it mimics the physiological conditions of healing wounds as shown by keratinocyte culture. ADSC secretome has recently been shown to accelerate wound closure, to enhance angiogenic response, and regeneration of epithelial gaps in mice skin wounds [25] and even to reduce scar tissue formation [26]. In addition, hair regeneration is positively influenced by ADSC secretome, possibly through up-regulation of insulin-like growth factor binding protein 1 and 2 (IGFBP-1/-2), MCSF, MCSFR and VEGF [27]. Growth factors such as VEGF, FGF and PDGF seem to play a key role and are also secreted in PL obtained from cord blood units. This is currently being investigated for potential clinical therapy in adequate substitution in artificial tears in corneal wounds [28].

Within this study, we could show that ADSC cultivated in cell culture medium supplemented with PL maintained stem cell characteristics such as typical stem cell pattern on protein level and multilineage differentiation capability. A systematic review by Guiotto et al. [29] demonstrates that PL is a suitable substitute in hMSC culture and even favourable in view of a clinical use. Lykov et al. [30] concluded that PL is as potent as FBS in stimulating proliferation, migration, and colony formation of MSC. Likewise, the same could be shown for other cell types, such as colorectal adenocarcinoma cells [31] and co-cultures of MSC and macrophages [32]. Kirsch et al. could further show that MSC in 3D hydrogels, mimicking the in vivo condition, proliferated best in medium supplemented with PL. This is in line with our study. ADSC viability varied significantly between PL and the 10% FBS group. Both PL groups showed a similar viability, which was significantly higher compared to the FBS group. While further studies also showed the superiority of PL as ADSC medium supplement [33], the exact dosage of PL remains unclear and varies between 2.5% and 20% [33,34]. While Kirsch et al. [33] observed the highest differentiation capacity of ADSC under 2.5% supplementation, Shih et al. [35] described an earlier osteogenic differentiation in 10% PL groups and overall similar results in 10% PL compared to 10% FBS. Other authors postulated best expansion rates in comparison with FBS supplementation at 10% PL [36] and 5% PL [37,38]. Thus, in this study, concentrations of 5% and 10% PL in comparison to 10% FBS were tested in ADSC culture. In our study, doubling the PL concentration from 5% to 10% could not further increase ADSC viability significantly. Five (5) % PL seems to be an appropriate supplement for stem cell cultures [39]. WST-8 and gene expression data demonstrated a higher viability and proliferation of cells in PL. For the analysis of cell cytotoxicity, further assays could be performed, such as analysis of LDH levels.

Not only the cell viability, but also *MKI67*, a known proliferation marker, was increased in PL-cultured ADSC, confirming a higher proliferation rate compared to the FBS cultures.

Overall, in all experimental groups, typical MSC surface markers were highly expressed, confirming no interference of PL in MSC characteristics. Negative markers, such as CD11b, CD19, CD31, CD34, CD45 and HLA-DR (Lin-cocktail), could not be detected, supporting the thesis of unaltered stem cell characteristics. A slight reduction in CD105 was noticed, although meeting the minimal criteria of MSC. This was described before by Riis et al. [10], who showed favouring of specific subpopulations in PL cultured ASC and altered expression of surface markers depending on the passage of used cells.

In all groups, ADSC could be successfully differentiated into adipogenic and chondrogenic lineages. Interestingly, adipogenic differentiation induced an earlier development of lipid vacuoles in PL groups compared to the FBS groups, most probably because of stimulating growth factors in PL. Conversely, other authors postulate a PL-induced suppression of lipocalin-type prostaglandin D2 synthase that positively controls adipogenic differentiation of MSC leading to a lower capability of adipogenic differentiation in PL-supplemented MSC compared to FBS-supplemented MSC, both meeting all common MSC criteria. This difference might be due to the fact that Lange et al. [40] used MSC derived from bone marrow, while we worked with stem cells from adipose tissue.

Both PL and FBS groups could be successfully differentiated into the chondrogenic lineage without visible differences. However, in previous studies, PL was described to enhance chondrogenic regeneration in vivo. In a clinical trial, autologous chondrocyte implantation was most successful under PL supplementation, showing faster regeneration and a lower inflammatory response compared to FBS supplemented groups [41]. Osteogenic differentiation of MSC was described to be enhanced in PL [42], however, in another study [43], no superior effects could be observed in osteogenic differentiation of equine bone marrow MSC under PL substitution, despite increased TGF-β1 concentration. In our study, ADSC that were cultured in PL, proliferated much faster in FBS supplemented medium, often leading to the detachment of cell layers during the differentiation period, hence not all groups could be analysed. For the same reason, osteogenic differentiation was not possible. Modifications to the protocol such as varying the cycle length, cell density, or using culture plate coatings, were not successful. However, all in all, it can be concluded that differentiation capability of ADSC is maintained or even enhanced when using PL as supplement in cell culture.

ELISA analysis of selected cytokines in PL-cultured ADSC secretome revealed that many markers were significantly increased in PL groups, e.g., VEGF, EGFR, FGF 4, MCSF, MSCFR and SCF. The VEGF family is a group of five factors, VEGF-A, -B, -C, -D and placenta growth factor (PIGF) known to regulate angiogenesis and lymphangiogenesis [44,45,46], but also to influence capillary permeability and stability of blood vessels. VEGF-A prolongs angiogenic signalling and increased in vivo wound closure [47,48]. VEGF-R2 is known as the main regulator of VEGF cellular responses [49] and is also significantly increased in the secretome of PL-cultured ADSC. EG-VEGF, also known as prokineticin, was observed to increase sprouting and vascular organization, and permeability in microvascular endothelial cells [50]. In the ovary, it was shown that it plays an overlapping role with VEGF in vasculogenesis [51]. In our study, PL enhanced the secretion of those three interacting proangiogenic factors. Enhanced proangiogenesis could increase keratinocyte cell culture growth and therefore skin regeneration. EGF-R is a receptor for EGF, a growth factor signalling a cascade of migration, proliferation, cytoprotection, and epithelial mesenchymal transition [52,53]. Clinical trials examined EGF as therapeutic agents [54] and concluded a promising new possibility in individual wound care. By increasing the EGF-R concentration, the effects of EGF must be supported. FGF-4 belongs to the fibroblast growth factor family, primarily identified as inducer of proliferation and differentiation in various cell types [55]. It was further found to be a key regulator in the migration of keratinocytes in skin defects [56]. Again, the rise of FGF-4 in PL groups indicates a beneficial role of PL-cultured ADSC in tissue repair. MCSF and MCSF-R are suspected to play a role in tumorgenesis, especially in glioblastomas [57], through induction of angiogenesis. The role of macrophages and their stimulation through cytokines are widely analysed, showing abnormalities in diabetic ulcers [58] and overall importance in tissue regeneration [59]. SCF and SCF-R are recruiters of mast cells, chemoattracting them to damaged wound sites and are not significant releasers of mediators, but recruiters of neutrophiles [60]. ICAM-1 is a cytokine known in rolling and transendothelial migration and cell adhesion. After activation, leukocytes bind to endothelial cells via ICAM-1 [61] and then transmigrate into tissues. As described earlier, inflammation is a substantial part of wound healing. The transmigration of leukocytes across vascular endothelia in processes such as extravasation and the inflammatory response is crucial in tissue regeneration. The expression of ICAM-1 was significantly increased in PL groups. ICAM-1 is suspected to upregulate its own expression through a positive feedback-loop, as well as upregulating RANTES [62], which was significantly increased in PL groups. RANTES is known as a classic chemotactic cytokine and induces expression of matrix metalloproteinases, especially MMP-2 and MMP-9, which are important for migration of cells into the site of inflammation [63]. This suggests a stimulation of inflammatory processes in cells supplemented in PL and is possibly beneficial for wound healing. MCP-1 is a small cytokine recruiting monocytes, dendritic cells and memory cells at the inflammation site [64]. At the site of wound healing, MCP-1 is also known to be predominantly produced in situ by tissue-resident macrophages and fibroblasts [65,66]. Fibroblasts, playing a key role in formation of granulation tissue, can be induced by MCP-1 to express TGF-ß and collagen [67]. MMPs are enhanced by human dermal fibroblasts through MCP-1 [68]. TNF-α is upregulated during the inflammatory phase. While low levels tend to promote wound healing by indirectly stimulating inflammation and increasing macrophage produced growth factors, high levels can be found in persisting wounds. One molecular mechanism is the enhancement of FGF through TNF-α, which promotes re-epithelialization [69]. Pro-inflammatory cytokines have been reported to have both inhibitory and stimulator of wound healing, this is why further investigations of the complex pathways need to be conducted. Moreover, this study could not detect significant differences between concentrations of interleukins, known pro-inflammatory cytokines. Using all cytokines in common is a major role in angiogenesis, differentiation, and migration in the wound milieu. The fact that PL induces significant higher concentrations of those proteins in ADSC secretome leads to the conclusion that PL-cultured ADSC could be a favourable substitute in wound repair. To understand the exact origin and influence of different cytokines, it is relevant to compare the composition of bare PL and PL-cultivated ADSC secretome via ELISA, which will be investigated in future studies.

Another aspect to be investigated is the effect of PL-cultured ADSC secretome on the production of reactive oxygen species. There is not yet data highlighting this interplay, but there is little data regarding PRF. PRF reduced the ROS release provoked by lipopolysaccharide (LPS) in RAW 264.7 cells (macrophages) and [70] therefore is supportive in periodontal disorders. Contrarily, other authors claim benefits of the activation of ROS. PDGF and VEGF are known to stimulate the amount of intracellular ROS by activation of NAPH oxidases complex, Martinotti et al. postulated that PL induces the increase of intracellular ROS [71]. In this study, VEGF was elevated in its concentration in PL-cultured ADSC secretome, therefore it could indicate increased levels of ROS. ROS production in ADSC treated with FBS and PL could be determined using the oxidation-sensitive fluorescent probe dichlorodihydrofluorescein diacetate (DCFH-DA) in future studies. It is known that platelets involved in wound repair and blood homeostasis release ROS to recruit additional platelets to sites of injury. Lastly, it remains unclear where the cut-off for excessive ROS production and therefore damaging effect lies. The aberrant accumulation of lipid ROS is associated with various acute and chronic conditions. In chronic wounds, an imbalance between radical-generating and radical-scavenging cellular systems might be crucial. A modern aspect of cell death is the concept of ferroptosis. Mechanisms controlling ferroptotic cell death have been investigated in recent years as therapeutic means for multiple pathologies [72]. Investigations of ferroptosis-inhibitors, e.g., ferrostatin-1, are currently performed regarding neurodegenerative and neuropsychiatric diseases; this could be examined in ADSC secretome as well.

EGF was used to optimize the effect of ADSC secretome for wound healing. In previous studies, EGF increased the VEGF and bFGF secretion in ADSC [16], positively influencing their proliferation and migration pattern. In this study, we could show that the secretome of ADSC cultivated in EGF supplemented medium induced increased viability and migration of keratinocytes. We decided to use EGF in a concentration of 100 ng/mL since in ADSC culture it was shown to influence the microenvironment effectively for tissue repair [73]. Another study demonstrated that EGF plays an important role in the proliferation and stem cell plasticity of ADSCs. Lower concentrations of EGF (5 ng/mL) in vitro enhanced proliferation of ADSC and affected their differentiation. It was shown that EGF supplementation may affect the pluripotent state of ASCs through downregulation of stem cell markers Oct4 and Lin28a [17]. ADSC secretome from ADSC pre-treated with EGF might be a worthwhile approach for epidermal differentiation, thus tissue engineering. For future investigations including a positive control, such as PDGF, could help to compare the effect of ADSC secretome with the therapy with single growth factors. To investigate the different impact and the effect of PL and ADSC on migration and viability, a co-culture of ADSC and keratinocytes and the sole use of PL on keratinocyte behaviour should be investigated in future studies. In future studies, a comparison between the traditional scratch assay and ORIS assay could help to highlight the effects of the ADSC secretome on keratinocytes in vitro. In addition, in vivo experiments are to be conducted in future studies, to gain more insight in further possibilities of clinical trials with PL cultured ADSC secretome.

Inflammation and hypoxia are closely connected on molecular, cellular, and clinical levels. One of the key regulators in this pathway is the hypoxia-inducible-factor (HIF) [74]. Hypoxia via the HIF pathway is known to enhance migration of dermal fibroblasts and keratinocytes as well as promoting proliferation [75]. Several genes, e.g., TGFB1, pro-collagen a1, PDGF and VEGF, are induced through stimulation of angiogenic processes [76]. Endothelial cells for example respond to hypoxia with a seven-fold increase in PDGF messenger RNA [77]. This study showed an increase in VEGF concentration in PL-cultivated ADSC secretome, supporting the thesis that a combination of hypoxia and a specialized secretome could be a useful clinical therapy in chronic wounds. Hypoxia combined with VEGF promoted ADSC proliferation and differentiation into endothelial cells trough demethylation of ephrinB2 [78]. Different studies showed increased levels of VEGF in hypoxia conditioned medium [79,80,81] to be advantageous in diabetic erectile dysfunction [82] and myocardial infarction in rats [79]. The key effect in pre-treatment with hypoxia in ADSC is enhancement of angiogenesis [79,80,82,83]. Pre-treatment with hypoxia significantly enhanced the paracrine effect of rat ADSCs [84]. Although no beneficial effects of hypoxia could be observed regarding keratinocyte migration and viability, its effect on angiogenesis would be highly supportive for wound healing, but was not analysed within this study. Therefore, hypoxia pre-treatment remains a useful support in optimizing ADSC secretome for clinical use.

On the other hand, anaerobic bacteria thrive under hypoxic conditions; half of all diabetic foot ulcers are infected. Since this is linked with worse outcome of such ulcers, often leading to amputations, antibiotic therapy is recommended [85]. In clinical settings, vacuum therapies are used to clean wounds and provide granulation stimulus through negative pressure and hypoxia. All in all, the benefits of hypoxia-induced changes are superior and even prevent collateral tissue damage through negative regulation of adaptive immunity. The clinical user must find a compromise between hypoxia induction and the overgrow of unwanted microorganisms.

One limitation of the study is the small number of patients. Especially in the ADSC secretome analysis, a high patient-to-patient variance was observed. Additionally, practical working can be limited by the increased cell growth in PL groups, although PL showed to be an equally usable substitute in ADSC culture. This led to a cell detachment in differentiation assays, causing experimental delays. Another major question to solve which was not addressed within this study and should be focused on in further studies, is the exact composition of PL, e.g., using PL fraction, protein separation and MS-analysis [33]. Another aspect for further clinical usage is the differentiation between autologous and allogeneic MSC origins. In this study, the focus was on autologous ADSC. As described by Mallis et al. [86], patients with medical preconditions, especially autoimmune diseases, are shown to exhibit impaired functional properties, compared to those obtained from healthy donors. In addition, donations of foetal MSC have better immunoregulatory functions. Chronic ulcers are not particularly autoimmune diseases, but it is open to discuss the donor quality, as patients with chronic wounds are more likely to be older and sick with various preconditions. This should be investigated in future studies.

Moreover, FBS is used worldwide und manufactured under strict standards. PL is also commercially available, but lacks a consistency of protocols for PL preparation, of standardized terminology and therefore treatment regimens [87].

## 4. Materials and Methods

### 4.1. Production of Platelet Lysate

PL was obtained from plateletpheresis donations from healthy volunteers at the Department of Transfusion Medicine and Haemostaseology, University Hospital of Erlangen, Germany. Platelet concentrates were stored at +22 °C ± 2 °C for 5 days under constant agitation until declared invalid for clinical transfusion. As all concentrates were intended to be transfused, all donors were healthy, met the minimal donor criteria and were aged 18 to 68 years. The gender was not recorded since all donors were anonymized in the process of possible transfusion. Only platelet concentrates containing 3 × 10¹¹ thrombocytes per 250 mL and less than 4 leukocytes per μL and less than 1000 erythrocytes per μL were used. One thousand (1000) mL of platelet lysate were obtained from 5 pooled thrombocyte concentrates each with a volume of 250 mL. Aliquots of platelet concentrates were frozen at −80 °C for a minimum of 48 h until further procession. Thawing was conducted using plasmatherm (Barkey, Cambridge, MA, USA) at 24 °C. A system from MacoPharma (Tourcoing, France) consisting of a freezing and centrifugation bag, a filtration, and a distribution kit were used for processing. The platelet concentrates then underwent two cycles of thawing at 37 °C and freezing at −80 °C for least 12 h and were then centrifuged for 30 min. A 0.65 μm filter was used for filtration by gravity. The bag containing filtered PL was recovered and connected to the 9-bag distribution kit. Aliquots of 25 mL were immediately stored at −80 °C until further usage. The entire procedure was performed under central temperature supervision systems and strictly closed systems.

### 4.2. Isolation and Cultivation of ADSC

ADSC were isolated from human abdominal fat tissue of six female healthy patients aged 33 to 52 (mean 40.5 ± 7.4 years). Human tissue collection was approved by the Ethics Committee of the Friedrich-Alexander-Universität Erlangen-Nürnberg (Germany) (Ethics number 264_13B) in accordance with the World Medical Association Declaration of Helsinki. Informed consent was obtained from all patients. About 20 mL fat tissue per patient was minced and digested with 0.1% collagenase type 1 (Biochrom GmbH, Berlin, Germany). The homogenous fat solution was centrifuged at 400× *g* for 10 min and the top fluid and fat layers were discarded. The cells were suspended in phosphate buffered saline (PBS) (Sigma-Aldrich, St. Louis, MO, United States), filtered (70 µm) and centrifuged (300× *g*, 10 min). Cell pellets were resuspended in minimal essential medium alpha (MEM α) (Biochrom GmbH) with supplements according to the following conditions: group 1: 5% PL, group 2: 10% PL, group 3: 10% foetal bovine serum (FBS superior, Biochrom GmbH); 1 × 10^6^ cells were seeded per T75 flask. One (1) % penicillin/streptomycin (Sigma-Aldrich) was added to the cell culture medium for the first two weeks. Cells were cultivated at 37 °C, 5% CO_2_. The medium was changed every 2–3 days and cells were split at about 90% confluence. Experiments were performed until passage 8.

### 4.3. Characterization of ADSC

Morphology of ADSC was checked on day 1 and 3 after isolation by microscopy (Olympus IX83, cellSens Software V1.16, Olympus Corporation, Tokyo, Japan). ADSC surface marker expression was analysed using flow cytometry (BD FACSVerse) (BD Biosciences, NJ, USA). In total, 1 × 10⁶ cells per group were incubated simultaneously according to the manufacturer’s specifications with: APC mouse anti-human CD73 (clone AD2), FITC mouse Anti-Human CD90 (clone 5E10) and PerCP-Cy™5.5 mouse anti-Human CD105 (clone 266) as positive markers; APC-Cy™7 mouse Anti-Human CD31 (clone WM59), CD34 PE (clone 581), CD45 PE (clone HI30), CD11b PE (clone ICRF44), CD19 PE (clone HIB19) and HLA-DR PE (clone G46-6) (referred to CD34, CD11b, CD19, CD45 and HLA-DR as Lin- markers) as negative control cocktail and mIgG1 PE (clone X40) and mIgG2a PE (clone GJ55-178) as a PE hMSC isotype negative control cocktail (all from BD sciences, NJ, USA). All CD markers are listed under Appendix A. The expression of positive markers should be >95% and was defined as the comparison of stained to unstained cells, with stained cells showing higher signal intensities than unstained cells. Negative markers should be expressed <2%. Performance QC and Assay Tube Settings set up using BD FACSuite CS&T Research Beads was performed according to the manufacturer’s protocol for diurnal control. Cells were discriminated according to the specifications of Wang et al. [88] and remain within the linear range of signal amplification. All experiments using flow cytometry were conducted following the guidelines by Cossarizza et al. [89], enabling reproducibility and including fluorescence minus one control. The average number of counted events was 60,000. Gating was performed on the forward and side scatter for viable cells and this gate used for the antibody stainings. Due to dispensability in terms of following experiments, this study refrains from measuring viability, signal-to-noise ratio, linear range and coefficient of variation.

### 4.4. Viability Assay

For comparing the cell viability of different experimental groups, a WST-8 assay was performed at time points 24 h, 48 h, and 72 h. In a 96-well plate, 2 × 10³ cells per well were seeded in triplicates in 100 µL cell culture medium according to the different experimental groups. After 24 h, 10 µL WST-8 solution (PromoCell GmbH, Heidelberg, Germany) was added and absorbance was measured after 2 h of incubation (37 °C) at 450 nm and 600 nm (background) using a microplate reader (Thermo Fisher Scientific, Waltham, MA, USA). One 96-well plate was used for each measuring point. Relative values (24 h = 1) were analysed.

### 4.5. Gene Expression of ADSC

RNA of the different experimental groups was extracted with the RNeasy Mini Kit with the corresponding QIAshredder Homogenizer (Qiagen, Hilden, Germany). QuantiTect Reverse Transcription Kit (Qiagen) was used for reverse transcription with DNase I incubation. Quantitative real-time PCR was performed using SsoAdvanced Universal SYBR^®^ Green Supermix (Bio-Rad Laboratories, Hercules, CA, USA) and a Light Cycler (Bio-Rad CFX96). All experiments were performed according to the manufacturer’s protocols. Tyrosine 3-monooxygenase/tryptophan 5-monooxygenase activation protein zeta (*YWHAZ*) served as housekeeping gene. Data analysis was performed using the 2^−ΔΔCT^ method. Samples were tested as triplicates. Primer sequences are specified in Table 1.

### 4.6. Differentiation of ADSC

For differentiation into the adipogenic and chondrogenic cell lineage, ADSC were cultivated in MEM α with supplements according to the experimental groups (group 1: 5% PL, group 2: 10% PL, group 3: 10% FBS). When reaching 90% confluence, the medium was changed to a differentiation medium. MEM α with supplements according to the different experimental groups served as control. Experiments were performed in triplicates.

For adipogenic differentiation, 2 × 10⁴ ADSC per well were seeded in 24-well plates and cultivated for 21 days in 500 µL MSC adipogenic differentiation medium I and II (ADMI I, II) (PeloBiotech GmbH, Planegg, Germany; ADMI I: basal medium, supplemented with 5% PL or 10% PL or 10% FBS, and insulin, glutamine, penicillin/streptomycin; 3-isobutyl-1-methylxanthine (IBMX), dexamethasone, rosiglitazone; ADMI II: basal medium, supplemented with 5% PL or 10% PL or 10% FBS, and insulin, glutamine, penicillin/streptomycin). ADSC were incubated in ADMI I for 3 days, followed by ADMI II for 24 h. This cycle was repeated 4 times. Differentiated ADSC were fixated in 4% buffered formaldehyde (Carl Roth GmbH + Co. KG, Karlsruhe, Germany), rinsed in PBS and stained with Oil Red O (Sigma-Aldrich) for 15 min. Pictures were taken by microscopy (Olympus IX83) at day 0, 7, 21 and after staining.

Chondrogenic differentiation was performed in a pellet culture system. In total, 2.5 × 10^5^ ADSC were transferred into a 15 mL tube in 0.5 mL chondrogenic differentiation medium (PeloBiotech GmbH) containing basal medium with ITS supplement (insulin, transferrin, sodium selenite), sodium pyruvate, proline, ascorbate, dexamethasone and TGF-β3. For creating cell pellets, tubes were centrifuged at 150× *g* for 5 min, and carefully placed into the incubator. The medium was changed 3 times a week. After 4 weeks, cell pellets were fixated overnight in 4% buffered formaldehyde. Cell pellets were embedded in paraffin. Five to ten slides per sample were cut and stained with alcian blue (Carl Roth GmbH & Co. KG, Karlsruhe, Germany). Slides were analysed by microscopy (Olympus IX83).

### 4.7. Analysis of Selected Cytokines of ADSC Secretome

ADSC were cultivated in cell culture medium according to the experimental groups until 90% confluence. Cell layers were washed with PBS and incubated in MEM α without supplements for 24 h at 37 °C, 5% CO₂. Supernatant was collected and immediately stored at −80 °C. Samples were analysed by ELISA Quantibody (Testing Service by RayBiotech Inc., Peachtree Corners, GA, United States) for Hepatocyte Growth Factor (HGF), Vascular Endothelial Growth Factor (VEGF), Vascular Endothelial Growth Factor-Receptor 2 (VEGF-R2), Endocrine Gland Vascular Endothelial Growth Factor (EG-VEGF), Endothelial Growth Factor Receptor (EGF-R), Fibroblast Growth Factor 4 (FGF-4), Macrophage Colony Stimulating Factor (MCSF), Macrophage Colony Stimulating Factor Receptor (MCSF-R), Stem Cell Factor (SCF), Stem Cell Factor Receptor (SCF-R), Intercellular Adhesion Molecule 1 (ICAM-1), Monocyte Chemoattractant Protein 1 (MCP-1) and Chemokine Ligand 5 (CCL5, RANTES).

### 4.8. Isolation, Cultivation, and Characterization of Keratinocytes

Abdominal skin tissue of one female healthy patient aged 41 was used for isolation of keratinocytes. Human tissue collection was approved by the Ethics Committee of the Friedrich-Alexander-Universität Erlangen-Nürnberg, Germany (Ethics number 264_13B), in accordance with the World Medical Association Declaration of Helsinki. Informed consent was obtained from the patient. Keratinocytes were isolated according to the protocol of the Epidermis Dissociation Kit, human (Miltenyi Biotec, Bergisch Gladbach, Germany). The skin was cut into 50 pieces with a diameter of about 4 mm. Ten (10) mL of Roswell Park Memorial Institute (RPMI) 1640 medium (Thermo Fisher Scientific) and 250 µL of Enzyme G (Miltenyi Biotec) were added to the tissue pieces. Samples were incubated for 18 h at 4 °C under continuous shaking using a tube rotator. Subsequently, epidermis and dermis could be separated manually. Five (5) mL of RPMI 1640, 125 µL of Enzyme P and 25 µL of Enzyme A (all from Miltenyi Biotec) were added to the epidermis samples and incubated at 37 °C for 60 min. The gentleMACS™ Octo Dissociator was used for sample shredding in gentleMACS™ C Tubes. Samples were centrifuged (10 min, 300× *g*), filtrated (70 µm) and diluted in complete keratinocyte growth medium (EpiLife^TM^ medium with Supplement S7; Thermo Fisher Scientific). Next, 1 × 10⁶ cells were seeded in collagen-coated (3 µg/cm^2^, collagen type I from rat tail, Cell Guidance Systems Ltd., Cambridge, UK) T75 flasks and cultivated at 37 °C, 5% CO_2_. A medium change was performed every 2–3 days and cells were split at about 70% confluence. Experiments were performed until passage number 2.

For characterization, a cytokeratin immunofluorescence staining was performed. Cells were washed with PBS, fixed in 10% formaldehyde for 15 min. Specimens were blocked with 5% goat serum (Alexa Fluor^®^ 488 Goat Anti-Mouse IgG1 (γ1) antibody) (Thermo Fisher Scientific, Waltham, MA, USA) for 30 min, followed by 1 h incubation with the primary antibody (monoclonal mouse anti-human cytokeratin clone MNF116) (Agilent, Santa Clara, CA, USA), which was diluted 1:100 with antibody diluent (ZUC025-100, Zytomed, Berlin, Germany). This antibody reacts to cytokeratin 5, cytokeratin 6, cytokeratin 8, cytokeratin 17 and cytokeratin 19. A negative control was performed using isotype control mouse IgG1 (negative control code X0931) (Agilent, Santa Clara, CA, USA). Cells were washed three times in TBS-T (Tris-buffered saline with Tween20) and counterstained with in 1:1000 distilled aqua diluted DAPI Staining Solution (Thermo Fisher Scientific, Waltham, MA, USA).

### 4.9. Preparation and Concentration of ADSC Secretome

For the preparation of ADSC secretome, ADSC were cultivated in T25 flasks in 3 mL MEM α supplemented with 5% PL until reaching 90% confluence. After washing with PBS, ADSC were incubated for 24 h at 37 °C, 5% CO_2_ in MEM α in 4 different conditions: group 1: normoxia (21% O_2_), group 2: hypoxia (1% O₂), group 3: normoxia (21% O_2_) with 100 ng/mL EGF, and group 4: hypoxia (1% O_2_) with 100 ng/mL EGF. As a control, MEM α was incubated without cells for 24 h at 37 °C, 5% CO_2_, 21% O_2_. Incubation was performed in technical triplicates. Supernatants from each group were pooled and 3-fold concentrated by centrifugation (Amicon Ultra-15 Centrifugal Filter device (Sigma-Aldrich) 4000× *g*, 30 min) and dilution with MEM α. ADSC secretome was immediately stored at −80 °C until further usage.

### 4.10. Effects of ADSC Secretome on Keratinocyte Viability and Migration

The effect of ADSC secretome on viability of keratinocytes was measured with the WST-8 assay (PromoCell GmbH). First, 2 × 10³ keratinocytes per well were seeded into a 3 µg/cm² collagen-coated (Cell Guidance Systems Ltd.) 96-well plate in triplicates in 100 µL complete keratinocyte growth medium. After 4 h, keratinocytes were cultivated in different types of ADSC secretome according to the groups 1–4, which was diluted 50:50 with complete keratinocyte growth medium. Medium was changed after 2 d and 4 d. At time points 1 d, 4 d and 7 d, 10 µL WST-8 solution was added and absorbance was measured after 2 h of incubation (37 °C) at 450 nm and 600 nm (background) using a microplate reader (Thermo Fisher Scientific). One 96-well plate was used for each measuring point.

For the migration assay, 3 µg/cm^2^ collagen-coated (collagen type I from rat tail, Cell Guidance Systems Ltd.) 96-well plates were prepared with stoppers using the Oris™ Cell Migration Assembly Kit (Platypus technologies, Madison, WI, USA) to create a central cell-free zone. Per well, 6 × 10^4^ keratinocytes were seeded in triplicates in 100 µL complete keratinocyte growth medium. After 4 h, stoppers were removed. Cell layers were washed twice with PBS and medium was added according to the ADSC secretome groups 1–4, diluted 50:50 with complete keratinocyte growth medium. Microscopic pictures were taken over a period of 24 h with an interval of 30 min (Olympus IX83).

### 4.11. Statistical Analysis

All experimental data are displayed as the mean of all independent experiments ± standard deviation (SD). The statistical analysis was performed with GraphPad Prism 8.4.3 (GraphPad Software Inc., San Diego, CA, USA). Comparisons between all groups were conducted using Kruskal-Wallis test, individual comparison between the groups was performed using Mann-Whitney U test. All experiments were performed in technical triplets using ADSC of 6 patients, except for ELISA quantibody, where due to technical issues only 5 patients were analysed. Differences were considered statistically significant at *p* ≤ 0.05 and highly significant at *p* ≤ 0.01.

## 5. Conclusions

With this study, we could show that PL is a suitable substitute in ADSC culture. Stem cell characteristics are preserved including stem cell marker expression and ability to differentiate. PL significantly enhanced ADSC viability and proliferation compared to FBS. Secretome from PL-cultured ADSC contained increased amounts of proteins, such as proliferation, differentiation or pro-angiogenetic factors compared to FBS-cultured ADSC, which stimulate keratinocyte migration and viability. EGF substitution and most probably also hypoxia for ADSC secretome production support these effects and finally can lead to better wound healing. Lastly, the study illustrates that ADSC secretome can effectively support tissue regeneration and is a promising tool in individual treatment of wound healing.

## Figures and Tables

**Figure 1 ijms-24-03522-f001:**
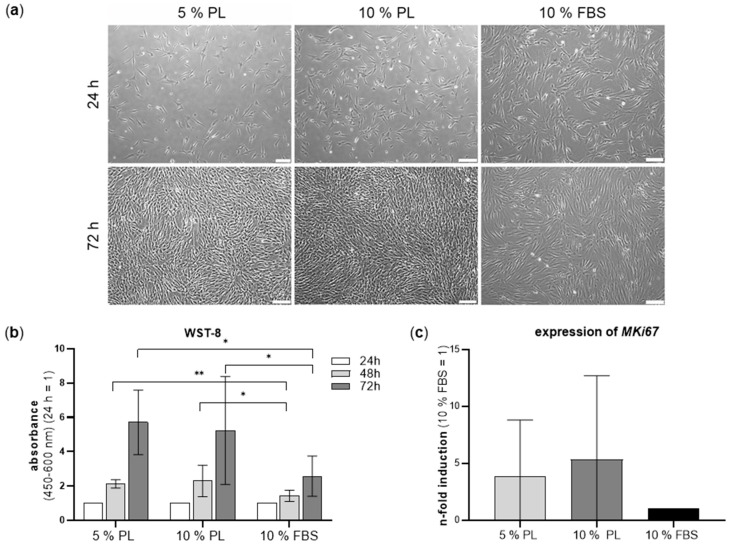
Morphology, cell viability and proliferation marker of ADSC cultivated in 5% PL, 10% PL and 10% FBS. (**a**) Morphology of ADSC after 24 h and 72 h. Scale bar 200 µm. (**b**) Bar graphs show the relative cell viability of ADSC in different groups at 24–72 h. (**c**) Real-time PCR of *MKI67* as proliferation marker represented as n-fold induction (10% FBS = 1). (*n* = 6, technical triplicates, * *p* ≤ 0.05, ** *p* ≤ 0.01).

**Figure 2 ijms-24-03522-f002:**
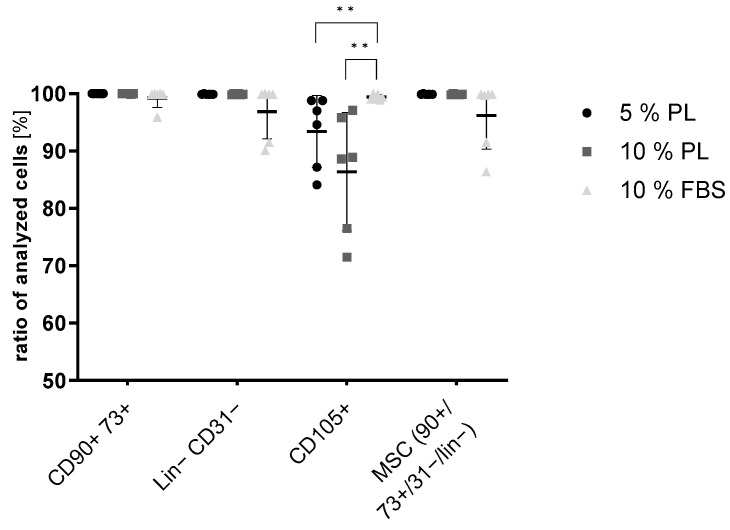
Flow cytometry of ADSC cultivated in 5% PL, 10% PL and 10% FBS: positive stem cell markers: CD90, CD73, and CD105; negative markers: Lin (CD34, CD11b, CD19, CD45, HLA-DR) and CD31 (*n* = 6, technical triplicates, ** *p* ≤ 0.01).

**Figure 3 ijms-24-03522-f003:**
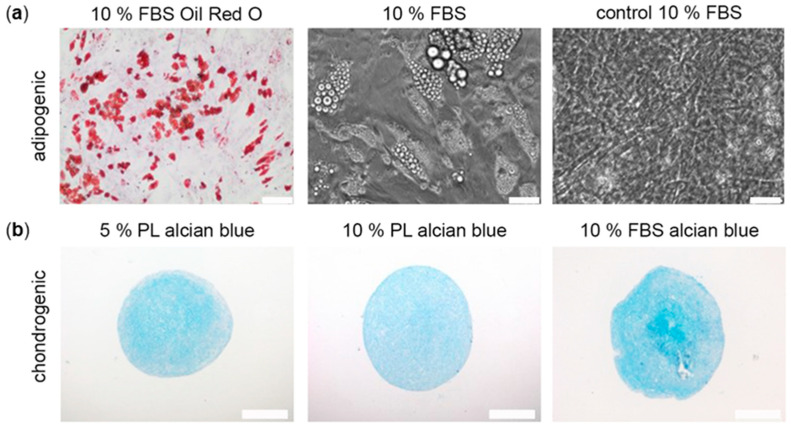
Effect of PL and FBS substitution on ADSC differentiation. (**a**) Representative images of FBS groups after adipogenic differentiation in Oil Red O staining and in phase contrast and of control groups in phase contrast. (**b**) Representative images of chondrogenic differentiated ADSC in alcian blue staining from day 28. Scale bar 200 µm.

**Figure 4 ijms-24-03522-f004:**
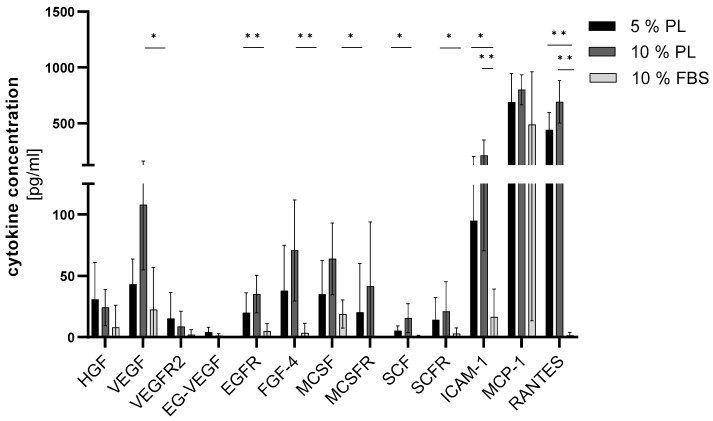
Effect of PL and FBS substitution in ADSC culture on cytokine concentration. Cytokine concentration (*y*-axis) of different proteins (*x*-axis) cohesive to proliferation and differentiation in wound healing shown as bar graph (*n* = 5, triplicates, * *p* ≤ 0.05, ** *p* ≤ 0.01 in comparison to FBS).

**Figure 5 ijms-24-03522-f005:**
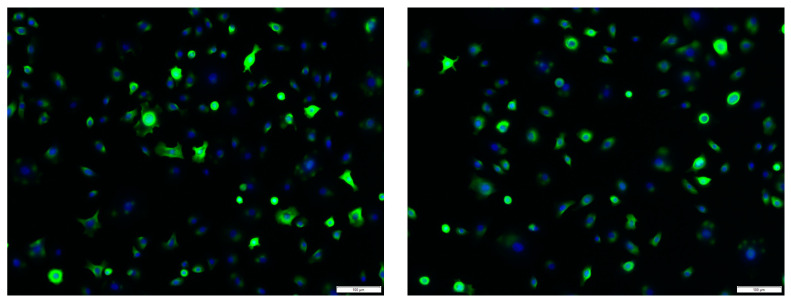
PAN-cytokeratin immunofluorescence staining of primary human keratinocytes (green). Counterstaining was performed with DAPI (blue). Scale bar 100 µm.

**Figure 6 ijms-24-03522-f006:**
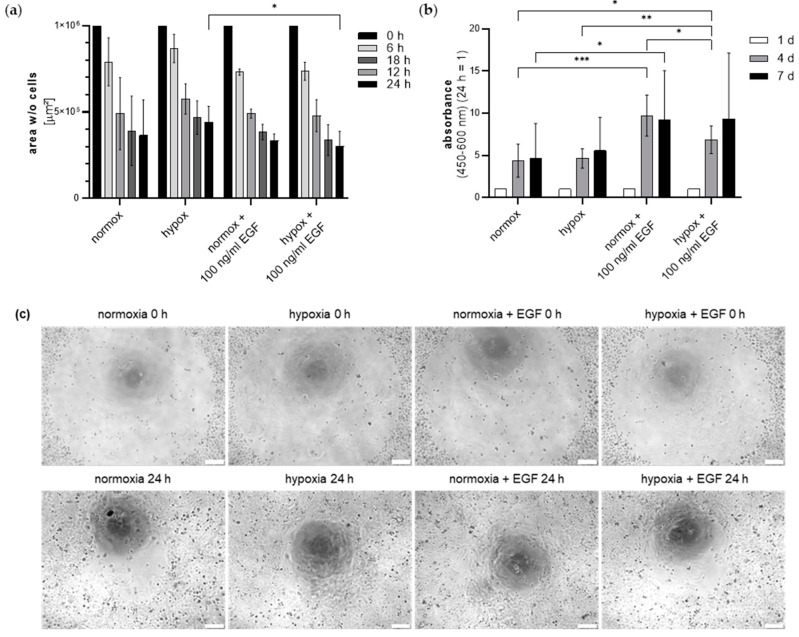
Effect of ADSC secretome from ADSC cultivated in normoxia or hypoxia with or without EGF supplementation on keratinocyte migration and viability. (**a**) Migration of keratinocytes depicted as cell-free area over 24 h. (**b**) Keratinocyte viability measured by WST-8 over 7 days. Values were calculated relatively (24 h = 1). (*n* = 3, technical triplicates, * *p* ≤ 0.05, ** *p* ≤ 0.01, *** *p* ≤ 0.005). (**c**) Representative images of keratinocyte migration after 0 h and 24 h. Scale bar 200 µm.

**Table 1 ijms-24-03522-t001:** Primer sequences.

Gene	Forward 5′->3′	Reverse 5′->3′
*YWHAZ*	ATGAGCTGGTTCAGAAGGCC	AAGATGACCTACGGGCTCCT
*MKI67*	TCGACCCTACAGAGTGCTCA	GTGGGGAGCAGAGGTTCTTC

## Data Availability

Data is contained within the article.

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
