# Peer review of "Secretome of Adipose-Derived Stem Cells Cultured in Platelet Lysate Improves Migration and Viability of Keratinocytes"

_ijms, 2023, doi:10.3390/ijms24043522_

Round 1
Reviewer 1 Report
Dear Authors the manuscript entitled " Platelet lysate improves cultivation of adipose-derived stem cells in wound healing therapy" represents a well designed and presented study in the field.
You can find my comments below:
1) In the section 2. Materials and Methods- 2.1 Production of platelet lysate, please provide as supplementary material the final obtained volume of PL and also the concentration of WBCs, RBCs, HgB and PLTs (before and after the thawing and filtration procedure).
2) In the section 2. Materials and Methods- 2.3. Characterization of ADSC. Please describe in detail the gating panel and the number of events that were obtained for the current measurements.
3) In the section 2.- 2.7. Cytokine analysis of ADSC secretome
It is better to perform a full proteomic analysis using mass spectrometry in order to have more details regarding the ADSC secretome. Otherwise please change the title of this section. The quantified cytokines represent only a small portion of the ADSCs secretome so the term is not acurate enough.
4) Before performing the kerotinocytes experiments, the authors have to characterize the keratinocytes e.g. immunophenotyping using flow cytometry.
5) Please include in your discussion the following publications
https://doi.org/10.37349/ei.2021.00010
- DOI: 10.3390/cimb44100303
Reviewer 2 Report
1) Why is the title based on wound healing, but the study design did not include an in vitro wound study model? An in vitro wound model to test this hypothesis would significantly add to the paper's contribution. For example a scratch assay (wound healing benchtop assay)
2) I would suggest placing the CD markers in a table to list those that were demonstrated or missing.
3) To my understanding, Platelet lysate is readily used in cell culture media to replace FBS. I am confused about the novelty of this based on the paper.
Reviewer 3 Report
Hermann and colleagues in their study deal with an interesting although not particularly innovative topic. Cell therapy for wound healing and in particular MSCs, exosomes, and platelet lysate is not an original topic, but, in my opinion is very interesting.
First of all, you are, in my opinion, lacking in the description of platelet lysate production and by this I mean: how many donors, health characteristics, age and gender.
Certainly the fact of not characterizing it in terms of growth factors, as described in the discussions, is an important factor in the discussion but the possibility of trying to make it as standardizable as possible is fundamental, above all because you are working with two elements that are too characterized by their source: PL and MSC. This makes your work hard to reproduce.
We know, and the large standard deviations that you have had also demonstrate it, that there is enormous variability when cultivating mesenchymal stem cells and even more so if they are cultivated with platelet lysate.
For this it goes, well stressed the characterization of the PL and how many batches of lysate were tested. It would be interesting to view the cell viability for each MSC analysed.
Then specify on which MSC the differentiation was performed.
Was secretome analysis performed on each MSC lines?
Thanks for your answer
Reviewer 4 Report
Dear authors
The work submitted is very interesting and embark on important factors regarding chronic wounds growth.
I would ask/recommend/suggest a few questions regarding the study conducted.
1. Chronic wounds are difficult to heal and among various factors, one of the leading causative reasons is excessive ROS production, which is further complicated by the involvement of ferroptosis (iron-induced oxidative stress caused by malfunctioning mitochondria). What do you think, would your platform serve as a potent ROS scavenger? if Yes, have you performed those experiments and tried to correlate your data with them?
2. Why the investigation of inflammatory biomarkers was not taken into account? besides, focus was given only to various growth factors? Do they also help in reducing the inflammatory phase time in chronic wounds?
3. Your study claimed (focused in intro section) that creating hypoxia at wound site will accelerate wound healing. But hypoxic conditions will also help improve anaerobic/acidophilic bacteria to complicate the wound. What would you say about it?
4. in M&M section, you have stated migration assay was also performed (line 245 to 252), but i dont forsee any of its results presented in the Results section. Please include those results.
5. Figure 1a, shows cell pictures taken at interval of 24 and 72 hours. Where are time 0 pictures? Please add.
Round 2
Reviewer 1 Report
Dear Authors,
You have successfully performed the revisions towards my comments. Good work!!!
Author Response
Dear Reviewer,
we thank you again fot taking the time to revise the manuscript and appreciate your input very much!
Reviewer 2 Report
Thank you for the creation of the table. I do appreciate the follow up on my suggestions/comments.
Author Response

(The authors gave the same response as above.)

Reviewer 4 Report
Thank you for your responses and subsequent revision.
Author Response

(The authors gave the same response as above.)
